

Leveraging PMF Time Series Characteristics from Multi-PAMS Measurements for
NMHC Source Investigation and Ozone Formation Insights
Duy-Hieu Nguyen[1], Hsin-Cheng Hsieh[1], Mao-Chang Liang[2], Neng-Huei Lin[3], Chieh-Heng Wang[4**], Jia-Lin Wang[1*]
[1] Department of Chemistry, National Central University, Taoyuan, 320317, Taiwan
[2] Institute of Earth Sciences, Academia Sinica, Taipei 115201, Taiwan
[3] Department of Atmospheric Science, National Central University, Taoyuan, 320317, Taiwan
[4] Center for Environmental Studies, National Central University, Taoyuan, 320317, Taiwan
*Correspondence to*:
[*]Jia-Lin Wang (cwang@cc.ncu.edu.tw)
[**] Chieh-Heng Wang (chwang1110@gmail.com)
**Key points**
•   High-resolution PMF time series resolved eight NMHC source factors across sites and seasons
•   Acetylene plumes in PAMS measurements served as an internal check on PMF performance
•   Highly reactive mixed sources drove ozone formation potential, even during moderate ozone
conditions
**Abstract**
Ozone pollution is a persistent concern in Taiwan's Kaoping region, where high industrial emissions
contribute to poor air quality. Although non-methane hydrocarbons (NMHCs) are key ozone precursors,
their sources and seasonal dynamics remain insufficiently resolved in this complex environment. This
study aimed to characterize NMHC sources and quantify their contributions to ozone formation across
seasons using high-resolution measurements from three Photochemical Assessment Monitoring Stations
(PAMS) combined with Positive Matrix Factorization (PMF). Eight distinct source profiles were resolved,
including petrochemical factors I & II, refinery, gasoline evaporation, mixed sources (vehicular/solvent),
acetylene, aged air mass, and biogenic emissions. The model effectively captured source-specific
signatures, notably the acetylene factor at Linyuan ($R^2 = 0.99$ with observations), serving as an intrinsic
check on PMF performance.   Source contributions varied by site and season, with the petroleum
industry as the dominant contributor (33-71%), especially at Xiaogang, while aged air mass (12-30%)
and mixed sources (2-29%) also played important roles. Despite petroleum dominance, highly reactive
species in the mixed source factor drove higher ozone formation potential (OFP). Seasonal and pollution-
level analyses revealed that even under moderate ozone conditions (MDA8 40–60 ppb), urban-industrial
emissions remained significant contributors to OFP. These findings advance understanding of source-
specific NMHC dynamics and highlight the value of multi-site, year-round monitoring for constraining
ozone precursor sources. The results underscore the need to prioritize controls on petroleum and urban-
industrial emissions to mitigate ozone in industrialized regions.
**Short summary**
Using year-round, high-frequency non-methane hydrocarbons measurements from three monitoring sites,
each with distinct source–receptor characteristics, this study applied the source apportionment model to
resolve eight sources and assess ozone-forming potential. Distinct acetylene plumes at Linyuan ($R^2 >$



0.99) provided an internal consistency check. Results reveal spatial–seasonal source variability and
highlight the roles of petroleum, mixed, and aged air sources in ozone formation management.
**Keywords:** Trajectory analysis, Regional transport, Source-receptor modeling, NMHC speciation
**1 Introduction**

43       Volatile organic compounds (VOCs) are key precursors in atmospheric chemistry and play a

substantial role in determining air quality (Guan et al., 2020; Guo et al., 2017). In the presence of nitrogen
oxides (NOx) and sunlight, VOCs undergo photochemical reactions that lead to the formation of ground-
level ozone ($O_3$), while their oxidation products contribute to secondary organic aerosol formation (Wu
et al., 2024; Mcfiggans et al., 2019). These processes are particularly intensified in rapidly industrializing
and urbanizing regions, where elevated emissions and complex atmospheric interactions amplify air
pollution (Zhang et al., 2022). The resulting decline in air quality poses substantial risks to both human
health and ecosystems (Ramírez et al., 2019; Xu et al., 2022). Given their complex roles and diverse
emission sources, a detailed characterization of ambient VOCs is essential to advance our understanding
of their source origins, chemical behavior, and contributions to air pollution.

53       Traditional VOC source analysis often relies on passive sampling techniques. Such as using

canisters at strategic locations to capture spatial and temporal variations in ambient concentrations
(Dumanoglu et al., 2014; Mo et al., 2015; Wang et al., 2018; Dong et al., 2024). While effective for
regional assessments, these methods lack the high temporal resolution required for detailed source
apportionment. To overcome this limitation, automated gas chromatograph (auto-GC) systems have been
developed, offering continuous, high-frequency VOC measurements that better support the analysis of
dynamic emission patterns (Wernis et al., 2022; Su et al., 2016; Chen et al., 2014; Henry, 2013). In
Taiwan, the Environmental Protection Administration (EPA) has established a Photochemical
Assessment Monitoring Stations (PAMS) network to provide real-time NMHCs monitoring data—a
subset of VOCs—forming a technical foundation for evidence-based air quality management and
research. These sophisticated monitoring techniques form a foundation for detailed source apportionment
studies (Chen et al., 2021; Gu et al., 2020; Languille et al., 2020), enabling more accurate evaluation of
emission contributions and their temporal variations. Initially, this network lacked a standardized
approach for identifying specific NMHC sources, with only one of the nine stations incorporating
targeted source-tracking capabilities in 2007 (Chen et al., 2014). However, since 2013, the network has
expanded to track emissions from major industrial zones, growing to a total of 15 stations to better
support the EPA's regulatory and scientific objectives (Nguyen et al., 2025).

70       The Kaoping region, located in southern Taiwan, is home to one of the country's largest industrial

complexes, situated near Kaohsiung Port—an important maritime trade hub in East Asia (Yeh et al.,
2022). This region faces persistent air quality challenges due to intensive industrial activity, including
petrochemical manufacturing, steel production, and power generation, which collectively represent
dominant sources of anthropogenic NMHC emissions (Huang and Hsieh, 2019). In addition to the
industrial sector, emissions from vehicular traffic further burden the area's air quality, including ships in
ports, urban development, biogenic activity, and long-range transport of pollutants (Chou et al., 2022;



Lin et al., 2007). Vehicular emissions, especially from gasoline combustion, contribute substantially to
ambient NMHC levels, as indicated by elevated concentrations of marker species such as isopentane and
n-butane (Shao et al., 2016; Mo et al., 2017). Meanwhile, the presence of isoprene from vegetation and
meteorological phenomena—such as sea-land breeze circulations and seasonal monsoons—complicates
the chemical transformation and transport of NMHCs across the region (Li and Wang, 2012; Cheng et
al., 2016). The local EPA authority has implemented emission control strategies targeting major industrial
zones to address these environmental concerns. These measures include stricter emission standards,
technical support for pollution reduction, and mandatory installation of factory gas recovery and
treatment systems. Despite these efforts, residents adjacent to industrial parks continue to express
concerns about air quality issues (Ko, 1996; Deng et al., 2022), suggesting that current measures have
not fully mitigated the impact of VOC emissions on surrounding communities. Given these complex and
intertwined emission sources, high-resolution NMHC monitoring from the PAMS network in the
Kaoping region offers a valuable database for investigating source contributions and understanding their
influence on regional air quality to improve air pollution management and air quality.

91        To quantitatively determine the contribution of different emission sources, receptor models such

as Chemical Mass Balance (CMB) and PMF are widely applied in atmospheric research (Su et al., 2019;
Na and Kim, 2007; Liu et al., 2008b; Lingwall and Christensen, 2007). Each approach has its strengths
and limitations. CMB requires detailed and well-characterized source profiles and is sensitive to
collinearity among input species, which can limit its applicability in complex source environments. In
contrast, PMF is a data-driven technique that extracts source profiles and their contributions directly from
ambient measurements, making it more flexible in situations where comprehensive source profiles are
unavailable but depend on a sufficiently large and high-quality dataset (Su et al., 2016). Recent advances
allow PMF to incorporate auxiliary information (e.g., known marker species or source profiles),
improving source identification accuracy (Yang et al., 2022). In this context, the continuous, speciated
NMHC data provided by the PAMS network create an ideal foundation for applying PMF to source
apportionment in the Kaoping region, where emission sources are diverse. However, identifying sources
is only one part of the broader air quality picture. To fully understand pollution dynamics in this region,
it is also essential to consider meteorological factors that influence the dispersion, accumulation, and
transport of NMHCs.

106       To address this, many studies have coupled PMF with the Conditional Probability Function (CPF),

which integrates PMF-resolved source contributions with wind direction data to infer the likely directions
of pollution sources (Pekney et al., 2006; Zhou et al., 2018; Pallavi and Sinha, 2019). CPF analyses
typically rely on the time series of factor contributions generated by PMF to calculate conditional
probabilities of wind directions associated with elevated contributions (e.g., using the 75th percentile
threshold). While CPF enhances the spatial interpretation of PMF outputs, it only partially leverages the
rich temporal information embedded in the PMF contribution time series. Despite the widespread
application of CPF, a key analytical gap remains: the temporal features of PMF-resolved factor
contributions—beyond their statistical distributions—are rarely used to further investigate source



behavior. These time series contain valuable insights, such as peak events, seasonal trends, or episodic

spikes in individual sources. Importantly, high-contribution events in specific time windows provide an

opportunity to integrate PMF with air mass back-trajectory analysis, linking these peaks to possible

upwind source regions. This approach enhances the interpretability of PMF beyond statistical association

and opens new avenues for spatiotemporal source identification.

This study aims to bridge this gap by applying PMF and CPF to identify and characterize NMHC

sources in the Kaoping region and analyze the temporal features of PMF factor contributions to guide

trajectory-based source tracking. By identifying high-contribution episodes for source factor and

performing back-trajectory analyses during those episodes, we can confidently infer the likely

geographic origins of the emissions. Compared to most PMF studies that rely on data from a single

receptor site and limited time resolution, our study leverages three PAMS sites with year-round hourly

data and distinct source–receptor characteristics, allowing for a more robust source apportionment and

regional representation. This multi-PAMS framework offers insights into NMHC dynamic emissions and

transport in one of the most industrialized areas in Taiwan.

## 2 Methodology

### 2.1 Study site description

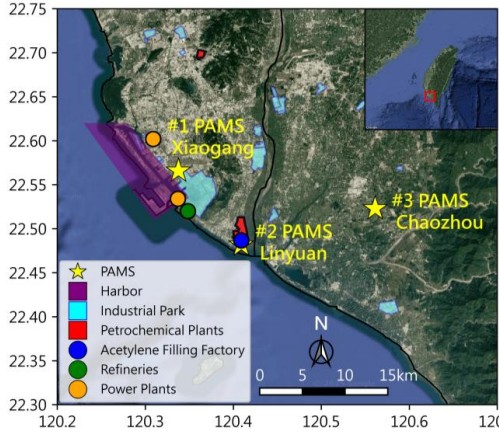

**Figure 1: Study region with marked industrial facilities. Base map from Google Maps (Map data ©2025 Google)**

This study focuses on source apportionment of ambient NMHCs in the highly industrialized

Kaoping region of southern Taiwan, approximately one-third the size of the New York metropolitan

region, based on measurements from three PAMS monitoring sites: Xiaogang, Linyuan, and Chaozhou.

The distinct locations of three sites and environmental settings provide a comprehensive view of source–

receptor relationships across the region. Xiaogang, located in southern Kaohsiung, is a mixed urban-

industrial area heavily influenced by emissions from multiple industrial activities. These include large

petrochemical complexes, steel mills, metallurgical processing plants, shipyards, and power stations. In

addition, maritime operations from Kaohsiung Harbor—one of the busiest ports in Asia—contribute





significantly to local NMHC levels. Linyuan, situated along the southwestern coast and downwind of
Kaohsiung's industrial corridor, is situated next to a large petrochemical complex frequently receiving
high-concentration plumes from nearby facilities under the influence of coastal meteorology. Chaozhou,
by contrast, is more inland and characterized by a predominantly agricultural and vegetative landscape,
with minimal presence of anthropogenic sources. However, due to its geographic position downwind of
the Kaohsiung industrial corridor, Chaozhou is susceptible to the regional pollution transported from
neighboring urban and industrialized areas. As such, it serves as a receptor site for background air quality
conditions with intermittent influence from upwind anthropogenic emissions. Together, the three sites
offer spatial and environmental contrasts that support robust source apportionment and allow for
evaluation of both local emission characteristics and regional transport dynamics in the Kaoping region.
**2.2 Data collection**
All three sites are equipped with PAMS, enabling high-temporal-resolution measurements of
speciated NMHCs (Fig. S1). Each station employs an automated gas chromatograph system (Clarus 500
GC, PerkinElmer), configured with dual flame ionization detectors (FIDs) and thermal desorption units
(TM-TD1, PerkinElmer). This TD-GC/FID setup allows for hourly quantification of 54 NMHC species
ranging from $C_2$–$C_{10}$. Ambient air is sampled at a flow rate of 15 mL/min over 40 minutes, yielding
approximately 600 mL of air per sample. Samples are pre-concentrated at –30 °C and desorbed at 325°C
for GC injection. A Deans switch system is employed to route lighter hydrocarbons ($C_2$–$C_5$) to an $Al_2O_3$
PLOT column (50 m × 0.32 mm i.d., 5.0 μm, Varian), while heavier compounds ($C_6$–$C_{10}$) are directed
through an uncoated column. Instrument calibration is conducted every five days using certified standard
gas mixtures containing the 54 target species (Spectra Gases, Branchburg, USA), ensuring data quality
with a measurement precision (1σ) maintained below 4% (Chen et al., 2014).
In addition to VOC measurements, this study incorporates hourly ancillary data from the Taiwan
Air Quality Monitoring Network (TAQMN), including ozone ($O_3$) and meteorological parameters such
as wind speed, wind direction, temperature, and relative humidity. This integrated dataset supports a
comprehensive analysis of photochemical activity, emission dynamics, and pollutant transport
mechanisms across the three monitoring environments: urban-industrial-port (Xiaogang), coastal-
industrial (Linyuan), and inland-rural (Chaozhou). After removing data affected by routine maintenance,
calibration events, or meteorological disturbances (e.g., typhoons, instrument shutdowns), the 2024
dataset includes 8,596 hourly samples, representing approximately 99% data coverage across the three
sites.
**2.3 NMHC source apportionment model**
PMF is a receptor-based modeling technique widely used for identifying and quantifying
contributions of pollution sources to ambient air quality. Originally developed by Paatero and Tapper
(1994) and refined in subsequent work (Paatero, 1997), PMF has been extensively applied to NMHCs
source apportionment in various urban and industrial environments (Guo et al., 2011; Zhang et al., 2015).
The method decomposes the observed concentration matrix (X) into two non-negative matrices: the
source contribution matrix (G) and the source profile matrix (F), along with a residual matrix (E). The



model accounts for measurement uncertainty and identifies latent factors representing individual sources,
each characterized by a distinct chemical profile and temporal pattern. Detailed mathematical
formulations are described in prior studies (Su et al., 2019; Han et al., 2023; Huang and Hsieh, 2019).
In this study, we applied the U.S. EPA's PMF 5.0 software to perform source apportionment of
NMHCs measured at the three PAMS sites in the Kaoping region. The input to the model consisted of
concentration and uncertainty matrices constructed from hourly NMHC measurements. Uncertainty ($U_{ij}$)
was calculated based on species concentrations ($X_{ij}$) and method detection limits (MDL) as follows:
$U_{ij}\sqrt{\left(0.5 \times MDL_j\right)^2 + \left(error\ fraction \times X_{ij}\right)^2}$         (1)
For concentrations below MDLs, the value was substituted with ½ MDL with uncertainty set at ⅚
MDL. Missing values were excluded from the input dataset to maintain model reliability. Species
selection was based on signal-to-noise (S/N) ratios and detection frequency. Specifically, species with
S/N ratios below 0.1 or lacking sufficient data across the three sites were excluded. This screening
process resulted in a final set of 22 out of the 54 measured species used for modeling. Model stability
was assessed by testing 3 to 8 factors, each with 100 independent runs using random seed initialization.
The optimal number of factors was selected based on the criteria of Q(robust)/Q(true) values approaching
1.0, reproducibility of factor profiles across runs, and the interpretability and physical plausibility of
identified source profiles.

**2.4 Directional analysis with CPF and trajectory modeling**

The CPF was employed to identify the likely directional origins of pollution sources by analyzing
the relationship between elevated PMF factor timeseries contributions and wind direction. First, source
contribution timeseries obtained from PMF output were used, with each factor representing a distinct
emission source (e.g., petrochemical, solvent usage, aged air mass). Wind speed and wind direction data
were integrated with PMF output to determine directional influence. For each PMF-resolved factor, the
70[th] percentile was the threshold to isolate the plume events.
$CPF_{\Delta\theta} = {n_{\Delta\theta}}/{m_{\Delta\theta}}$         (2)
CPF was computed as the ratio of the number of times the factor contributions exceeded the
threshold within a given wind sector ($n_{\Delta\theta}$) to the total number of valid observations in that sector ($m_{\Delta\theta}$).
Wind direction was divided into 16 equal intervals (22.5° per sector) to ensure robust analysis. Higher
CPF values in specific wind sectors indicated stronger contributions from sources in that direction. CPF
plots were generated for each PMF factor to visualize dominant source directions and assess consistency
with known emission source locations, meteorological patterns, and local topography.
Our developed trigger back-trajectory is employed to further utilize the time series features from
PMF factor contribution, particularly their episodic spikes or peak events. Unlike traditional Lagrangian
models such as HYSPLIT, which simulate long-range air mass transport using synoptic meteorological
fields, the trigger back-trajectory model is optimized for short-range, near-surface pollution episodes
using high-resolution wind observations from receptor sites. Each trajectory is then visualized and
mapped using GIS tools layered onto spatial imagery via the Google Maps API. By aggregating multiple





trajectories associated with similar high-concentration events, a trajectory ensemble analysis is
conducted to identify convergence zones, which are likely source regions. This hybrid approach improves
the spatial interpretability of PMF results by complementing factor profiles and CPF with spatiotemporal
back-tracing, providing a robust framework for identifying not just what the sources are, but when and
where they likely originated.
**3 Results and discussion**
**3.1 PAMS data overview**
Leveraging three-PAMS site in a region with heavy industrial loading, this study captures the
spatial variability of source dynamics, providing a clearer picture of how NMHC levels are shaped. The
yearly mean concentrations of NMHC—calculated from 54 measured species—exhibited significant
spatial heterogeneity across the sampling sites. As illustrated in Fig. 2, the level varied considerably, with
Linyuan recording the highest average concentration ($26.19 \pm 41.65$ ppb), followed by Xiaogang ($16.25
\pm 13.65$ ppb) and Chaozhou ($9.25 \pm 8.29$ ppb). Linyuan stands out as an NMHC emission hotspot ($26.19
\pm 41.65$ ppb), likely due to its dense petrochemical infrastructure, where variations in industrial
operations and meteorological conditions drive substantial fluctuations in emission levels. ($\pm 41.65$ ppb).
In contrast, with nearly three times lower NMHC levels, Chaozhou reflects a predominantly agricultural
setting characterized by more stable emissions with low standard deviation. While Xiaogang occupies
an intermediate position, with an average concentration of $16.25 \pm 13.65$ ppb, telling a story of mixed
urban-industrial character where both vehicular emissions and industrial activities contribute to ambient
NMHC levels, with its moderate variability reflecting the complexity of its urban ecosystem. A
comparative analysis with other urban environments reveals that NMHC concentrations in this study
were generally lower than those reported in Guangzhou (42.74 ppb), Wuhan (34.65 ppb), Chengdu (41.8
ppb), and Beijing (29.12 ppb), (Li et al., 2022; Hui et al., 2018; Zou et al., 2015; Song et al., 2018). This
overall lower presence of NMHC may suggest effectiveness in emission control, supported by stringent
regulations, improved fuel quality, and industrial emission standards. Additionally, meteorological
factors, such as higher wind speeds at Xiaogang and Chaozhou, likely contribute to dilution and
dispersion, further shaping the observed NMHC distribution.
**3.2 NMHC compositions**
While NMHC concentrations provide a broad picture of emission intensities, a more detailed
understanding of their pollution requires an analysis of their chemical composition. The NMHC profiles
at Chaozhou, Linyuan, and Xiaogang exhibited distinct characteristics, reflecting their diverse emission
sources and atmospheric processing (Fig. 2). Across all sites, 54 NMHC species were quantified and
categorized into four major groups: alkanes, alkenes, aromatics, and alkynes (ethyne).
Alkanes dominated the NMHC composition at Chaozhou and Xiaogang, comprising 57% and 60%
of total NMHC, respectively, while accounting for only 33% at Linyuan. This alkane dominance is
consistent with previous findings in Asian cities (Zhang et al., 2020; Song et al., 2020), reflecting their
long atmospheric lifetimes and broad emission sources, including hydrocarbon processing and gasoline-
related activities. Ethane, propane, and n-butane alone contributed 34–41% of total concentration across





three sites, with enhanced levels of n-butane and isobutane particularly evident at Xiaogang and
Chaozhou. In contrast, Linyuan exhibited a distinctly different chemical profile, with ethyne (acetylene)
making up 40% of its total NMHCs, substantially higher than at Chaozhou (15%) and Xiaogang (2%).
This substantial presence of acetylene indicates localized, intense anthropogenic activities. Notably,
Linyuan recorded the highest yearly mean NMHCs (26.19 ± 41.65 ppb). However, this ranking is
primarily driven by its elevated acetylene levels. If acetylene is excluded, Linyuan would rank second,
after Xiaogang, highlighting the disproportionate influence of a single pollutant species on the site's
overall NMHC burden.

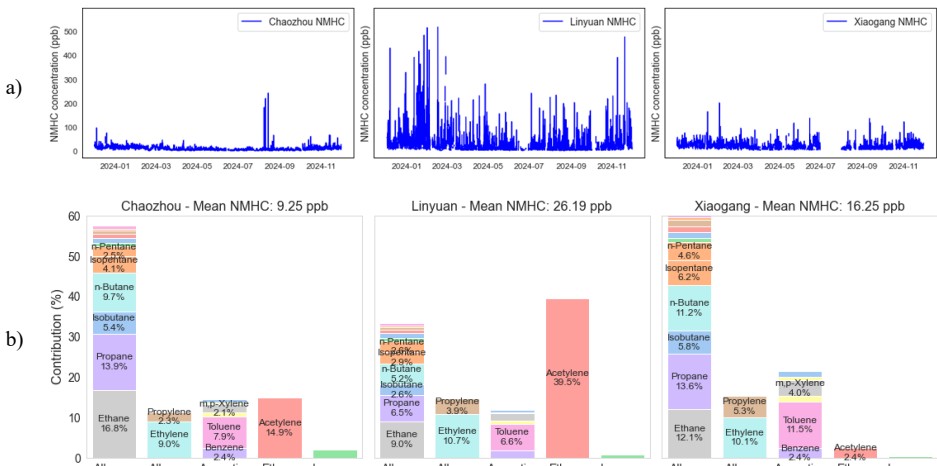

**Figure 2: Time series of total NMHC concentrations and mean composition of NMHC groups at**
**Chaozhou, Linyuan, and Xiaogang in 2024. (a) hourly variations, highlighting temporal patterns**
**and episodic peaks at each site. (b) averaged NMHC concentrations (ppb) and percentage**
**contributions of individual compounds within major chemical groups**
Aromatic compounds formed the second-largest group at Xiaogang (21%), followed by Chaozhou
(14%) and Linyuan (12%). The predominance of toluene and m,p-xylene across all sites aligns with
findings from previous studies in Shanghai (Zhang et al., 2018) and Xi'an (Song et al., 2020), indicating
contributions from solvent use, paint application, and industrial processes. The notably higher toluene
proportion at Xiaogang suggests significant emissions from solvent-related industries, such as coating
and painting. Alkenes accounted for a relatively stable fraction across sites (11% at Chaozhou, 15% at
Linyuan and Xiaogang), with ethylene being the dominant species. Given that ethylene is primarily
emitted from combustion and petrochemical activities, its consistent presence underscores the role of
anthropogenic sources in shaping the NMHC composition.
These pronounced differences in NMHC chemical profiles highlight the spatial heterogeneity of
emission sources and atmospheric processes across the study area. The alkane-rich profiles of Chaozhou
and Xiaogang contrast sharply with the ethyne-dominated composition at Linyuan, illustrating how
industrial activities significantly influence ambient NMHC signatures. Understanding these chemical
distinctions is crucial for designing targeted pollution control strategies tailored to the unique emission
characteristics of each urban environment.



### 3.2 Resolved Source profiles

As mentioned before, some measured NMHCs were excluded from PMF analysis due to their greater volume of data below MDLs. Consequently, the numbers of NMHC species input into the model for source apportionment were 22 species in 2024, which accounted for 88%, 91%, and 93% of the total NMHC concentration at Xiaogang, Linyuan, and Chaozhou, respectively. Based on the model used, there were eight distinct sources of resolved factors (Fig. S2), which are petrochemical I (Petro I), petrochemical II (Petro II), refinery, gasoline evaporation, mixed sources (Mixed), photochemical aged (Aged air mass), acetylene, and biogenic. Consistent source profiles were observed across seasons and at all three monitoring sites, underscoring the robustness of the PMF results and confirming the dominant contribution of specific source factors throughout the study area.

#### 3.2.1 Common sources

*a) Petrochemical factors*

The PMF analysis revealed a strong presence of ethylene and propylene, both of which are key raw materials in the petrochemical industry (Leuchner and Rappenglück, 2010). Ethylene is the most important feedstock in the synthetic organic chemical manufacturing industry, serving as a building block for a wide array of chemicals for making plastics, antifreeze solutions, and solvents. Similarly, propylene is a critical precursor in producing various petrochemical products. Our results are in agreement with those of Xie and Berkowitz (2006), who reported that ethylene and propylene were the main NMHCs emitted from petrochemical emissions. The factor represented by a single dominant compound of ethylene and propylene can be referred to here as the Petro-I and II (Fig. 3). Ethylene and propylene did not appear in proportion despite their shared industrial origin due to the spatial heterogeneity in emission sources across the large campus of the petro-complex at the studied sites. Notably, in summer, the Petro-I profile becomes especially pronounced at Xiaogang, characterized by increased ethane and propane as by-products of cracking operations (Thiruvenkataswamy et al., 2016; Pedrozo et al., 2020). They are prone to fugitive emissions or evaporative losses, particularly from storage tanks at elevated temperatures.

The PMF-resolved time series contribution for Petro-I and Petro-II reveals episodic spikes, with no clear pattern of increase or decrease during weekends (Fig. S3), suggesting these events may be associated with perennial petrochemical processes with constant fugitive emissions. Linyuan consistently exhibits the highest contributions for both Petro-I and II during winter and, to a lesser extent, spring and fall, followed by a distinct decline in summer. In contrast, Xiaogang's Petro-I contributions rise notably in fall and remain elevated into the winter with a lesser extent in spring, though Petro-II activity there is less pronounced. Chaozhou records consistently low contributions for both factors throughout all seasons, indicating minimal influence from these industrial sources (Fig. S3, green line). Spatial analysis of the preferred source directions for Petro-I and II further underscores site-specific differences. At Xiaogang, the dominant influence comes from northerly winds in fall, spring, and winter (Fig. 4, red line), while stronger southerly winds are observed in summer and peak in fall, emphasizing the role of prevailing winds in the Kaohsiung area. It highlights the complexity of source contributions and the importance of meteorological conditions in modulating observed concentrations. This pattern aligns with the spatial



distribution of active petrochemical facilities in Kaohsiung City, as shown in Fig. 1, and is reflected in
the dominant source directions in Fig. 4. For Linyuan, the prevailing source direction is from the
northwest, consistent with the site location downwind of Linyuan industrial areas (Fig. 4, orange line).
*b) Refinery factor*
The refinery-related emissions were primarily composed of C3–C5 alkanes, including propane,
isobutane, n-butane, isopentane, and n-pentane (Fig. 3). While n-butane and n-pentane are well-known
markers of gasoline evaporation, they are also associated with emissions from petroleum refining
processes (Wei et al., 2016). The separation of refinery and petrochemical sources in PMF analyses has
also been documented in previous studies (Kim et al., 2005; Buzcu and Fraser, 2006; Dumanoglu et al.,
2014; Chen et al., 2019) and in CMB modeling (Scheff et al., 1989). The PMF-resolved time series
contributions are relatively stable and exhibit consistent patterns across most sites and seasons, except
for a notable increase at Xiaogang during summer (Fig. S3). Occasional peaks in the PMF-resolved time
series at Xiaogang and Linyuan suggest the presence of episodic events or localized influences, likely
associated with butanes, which are the most dominant contributors to this factor. Yet, there is no clear
indication of a long-term trend or significant weekend effect, highlighting the ongoing and process-
driven nature of refinery emissions. Chaozhou consistently registers the lowest refinery factor
contributions in all seasons, likely due to its inland downwind position and limited proximity to major
refinery facilities. Directional analysis reveals that the preferred source direction for this factor is
southeast (SE) for Xiaogang and northwest (NW) for Linyuan, reflecting the locations of refinery
facilities relative to each monitoring site (Fig. 4).
*c) Gasoline evaporation*
Isopentane and n-pentane are recognized tracers of gasoline evaporation (Zheng et al., 2018; Chang
et al., 2022), and showed notably high contributions in our PMF results, indicating a strong influence
from evaporative sources (Fig. 3). This observation aligns with earlier studies that reported elevated
levels of light alkanes as characteristic of gasoline-related sources (Wang et al., 2015; Liu et al., 2008a).
In the Kaoping region, these species are primarily attributed to fugitive emissions from gasoline handling
and storage, such as the floating roof tanks, which are abundant in the refinery plants. The PMF-resolved
time series of gasoline evaporation at all three sites—Xiaogang, Linyuan, and Chaozhou—are generally
low and stable across all seasons (Fig. S3). Occasional minor peaks are observed, with Xiaogang and
Linyuan exhibiting slightly higher values than Chaozhou at times; however, there are no significant or
persistent episodic events. This pattern underscores the nature of fugitive pollution associated with
gasoline evaporation in industrial areas, which remains a minor and relatively steady contributor to
ambient NMHC levels year-round. Analysis of CPF values further reveals distinct spatial and seasonal
patterns. At Linyuan and Xiaogang, higher CPF values for gasoline evaporation are consistently
associated with winds from the N, NW, and SE, SSE directions, respectively, suggesting that elevated
concentrations are most likely to occur under these prevailing wind conditions (Fig. 4). Winter and spring
generally display slightly higher CPF values than summer and fall, particularly at these industrial sites,
reflecting seasonal variations in atmospheric transport or emission dynamics. In contrast, Chaozhou



consistently exhibits the lowest CPF values and NMHCs abundance across all seasons and wind
directions, indicating minimal impact from fugitive gasoline evaporation emissions. This is likely
attributable to the absence of industrial facilities and the rural setting at the site, as well as its downwind
position relative to Xiaogang and Linyuan, with the main influx of pollution arriving from the west and
west-southwest directions.
*d) Mixed factor*
In this study, distinguishing between vehicular and solvent-related emissions proved challenging,
primarily because Kaohsiung City hosts large-scale petrochemical complexes, shipyards, and harbor
facilities. Our PMF analysis was unable to resolve these sources as distinct factors but instead
appropriately captured this complexity under a single, mixed source factor (Fig. 3). This likely reflects
the complex emission environment of the region and possibly the proximity in distance of these source
types, where vehicular activities and solvent usage co-occur, contributing overlapping NMHC species
such as benzene, toluene, ethylbenzene, etc (Chang et al., 2003). A major limitation in separating these
sources is the absence of key tracers, such as methyl tert-butyl ether (MTBE), as an indicator for vehicle
exhaust or evaporation (Rubin et al., 2006). Without such specific markers, and given the shared NMHC
species between these sources, the PMF model likely grouped them under a single mixed profile.
As a result, the mixed source factor exhibits characteristics of both vehicular and solvent usage
emissions. Cai et al. (2010) reported that ethylene and propylene are major species produced by internal
combustion engines, and both were present in our mixed factor. Additionally, acetylene—a known tracer
of incomplete combustion—along with toluene, benzene, and m,p-xylene, typify vehicular emissions,
further supporting the presence of traffic-related contributions (Liu et al., 2008a; Nelson and Quigley,
1984; Xu et al., 2017; Baker et al., 2008). Cyclopentane, 2-methylpentane, and methylcyclopentane are
primarily markers of fuel evaporation and unburned fuel emissions from vehicles, rather than direct
combustion products. Still, they can also occur in vehicle exhaust due to incomplete combustion or as
unburned hydrocarbons. Isoprene may also be present in vehicle exhaust, especially under conditions
of incomplete combustion (e.g., cold starts, older engines, or engines lacking effective emission controls)
(Park et al., 2011; Nakashima et al., 2010; Zou et al., 2019) or from tire degradation (Jung and Choi,
2023). Moreover, the increased contribution of heavier aromatic compounds within the mixed factor
shown in Fig. 3 suggests a potential influence from diesel truck exhaust (Wang et al., 2024). The
frequent presence of heavy-duty diesel trucks transporting steel, chemicals, machinery, and other goods
was a common sight on the roads of Xiaogang and Linyuan, serving as the logistical backbone for
multiple heavy industries in the region. Simultaneously, the presence of compounds such as toluene,
ethylbenzene, m,p-xylene, n-hexane, and 1,2,4-trimethylbenzene in the same factor points to significant
solvent usage (Wu et al., 2016; Shen et al., 2018; Shao et al., 2016). These species are widely used in
paints, adhesives, coatings, cleaning agents, and chemical manufacturing processes common in
industrial zones like Xiaogang and Linyuan. Notably, toluene is a dominant solvent species (Bari and
Kindzierski, 2018).





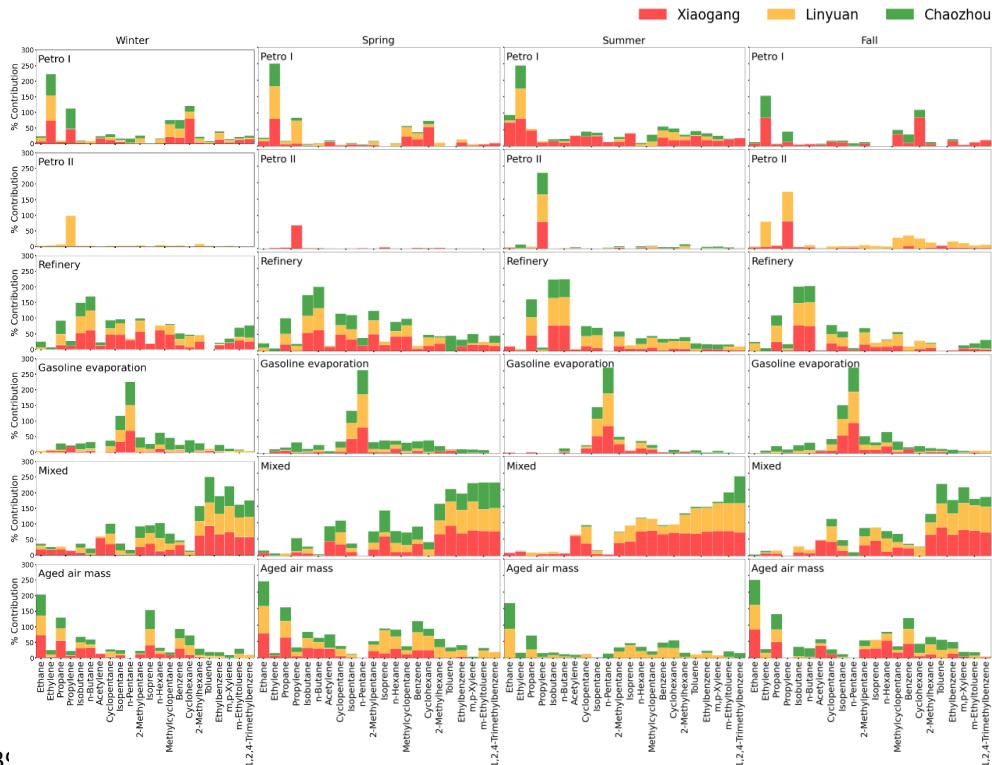

39.

**Figure 3: Summary of common source profiles of NMHC at the three sites in 2024**

By observing PMF-resolved time series (Fig. S3), the mixed factor exhibits strong seasonality, with its contributions peaking in winter, reaching the lowest levels in spring and summer, and rising again in the fall. Xiaogang and Linyuan consistently display higher values than Chaozhou, especially during peak winter and late fall, highlighting the greater influence of industrial and traffic activities near these sites. While generally lower and more consistent, Chaozhou still shows some variability, indicating a minor but persistent influence from mixed sources. Interestingly, the mixed source factor often demonstrates a weekly concentration pattern characterized by elevated weekday levels and lower concentrations on weekends (Baidar et al., 2015; Pollack et al., 2012), further supporting the contribution from traffic-related emissions.

Across all seasons, CPF values for the mixed factor remain relatively stable and uniform, suggesting that the contributing sources are predominantly local and affected by winds from multiple directions. However, during the fall, there is a noticeable increase in CPF values from the NW and NNW sectors, corresponding with the onset and intensification of the northeastern monsoon (Fig. 4). This shift results in more frequent and stronger winds from these directions, enabling enhanced transport from upwind or regional sources and producing more pronounced CPF values from NW and NNW. Meanwhile, the elevated time series contributions observed during winter are likely due to accumulation in the atmosphere, driven by a lower mixing layer height, which promotes the buildup of locally emitted pollutants (Fig. S3).



*e) Aged air mass factor*

The aged air mass factor was characterized by a dominant presence of low-reactivity, long-lived (NMHCs), including ethane, propane, acetylene, and benzene (Fig. 3). These compounds are sufficiently stable to survive long-range atmospheric transport due to their relatively slow reaction rates with hydroxyl (OH) radicals. The atmospheric chemical lifetimes of these VOCs range from several days to months, allowing them to persist in the environment and become enriched over time (Atkinson and Arey, 2003; Lau et al., 2010). Such compositional features suggest that the aged air mass is primarily influenced by secondary and transported sources rather than recent local emissions. Ethane, in particular, often dominates this factor due to its long atmospheric lifetime, and its strong presence aligns with findings from previous studies (Chen et al., 2010; Li et al., 2015). Similarly, benzene and acetylene are frequently observed in aged air masses, consistent with the findings of Wu et al. (2016), suggesting regional transport rather than local fugitive emissions as their primary source.

The PMF-resolved time series of the aged air mass factor reveals distinct seasonal trends. Contributions from aged air mass rise in late fall, peaking during winter, most notably at Xiaogang and Linyuan (Fig. S3, the red and orange lines). This elevated influence persists into spring before gradually declining, a pattern that closely mirrors the strengthening and subsequent weakening of the northern monsoon. While Chaozhou is located further inland, its aged air mass time series remains comparable to those at Xiaogang and Linyuan, though slightly lower in magnitude. This suggests that Chaozhou is still significantly influenced by long-range transport and potentially nearby sources from the vicinity areas. These spatial differences underscore the critical role of seasonal wind direction and site location in shaping the transport and accumulation of aged air masses across the sites.

Analysis of CPF values further supports these findings, with generally higher values observed from the NW to N wind sectors (Fig. 4), consistent with the prevailing northern monsoon during late fall through spring. Xiaogang and Linyuan display pronounced CPF peaks in these directions, reinforcing their susceptibility to long-range transport and downwind positioning during the monsoon. This pattern aligns well with the elevated time series signals observed at these sites during winter and early spring.

By contrast, Chaozhou exhibits lower CPF values and less direct influence from the northern monsoon. Instead, the CPF values at Chaozhou are enhanced under western wind conditions, suggesting that local factors, such as inland positioning and the influence of sea–land breeze circulation, modulate its exposure to aged air mass transported from Xiaogang and Linyuan. Fig. S4 further supports this observation as it shows the wind profile during day & night in alignment with the sea-land breeze pattern at Kaohsiung. Taken together, these patterns highlight the interplay of regional transport, seasonal meteorology, and site-specific geography in determining the concentration and source influence of aged air masses across the study area



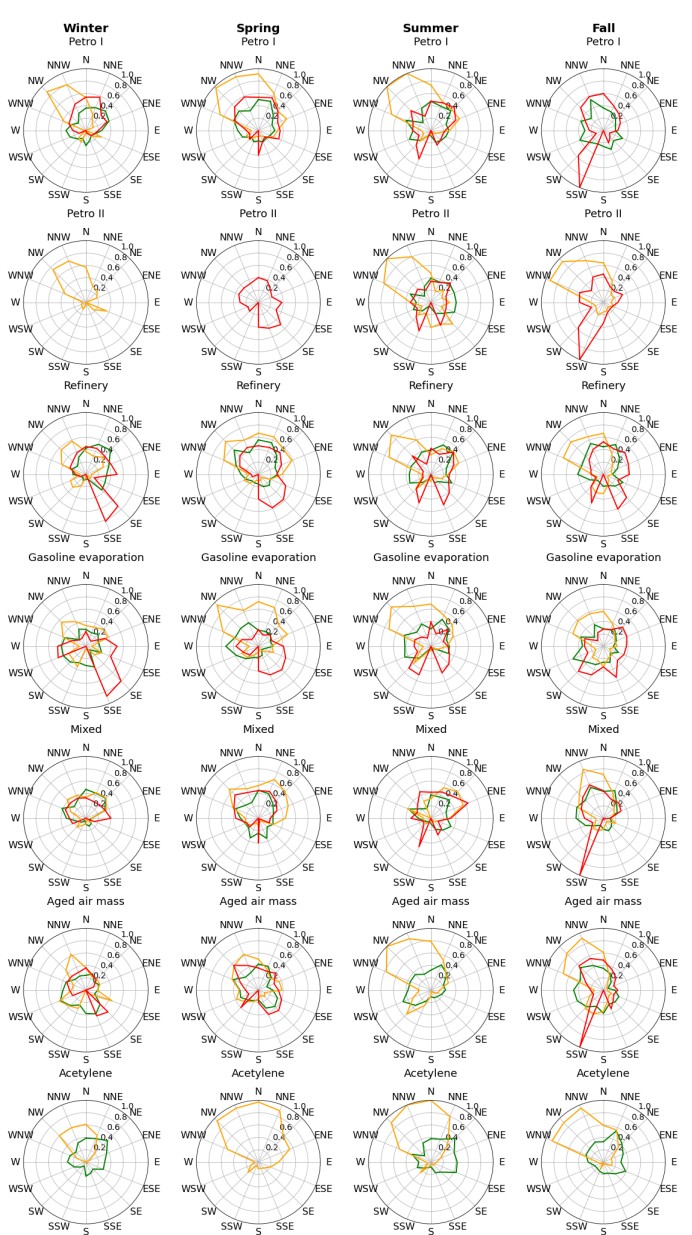

**Figure 4: CPF results viewing the direction for the highest 30% of factor contribution**

### 3.2.2 Distinct sources

*a) Biogenic factor*

There are distinct sources that bind closely with a specific site. The PMF results at Chaozhou identified a biogenic factor dominated by isoprene, with contributions present in both summer and fall (Fig.5). Notably, the time series and diurnal profiles (Fig. S5a,b) indicate that isoprene levels are





consistently higher in summer than in fall, both in terms of magnitude and variability. This seasonal
difference is consistent with the known emission behavior of isoprene, which is highly sensitive to
ambient temperature and solar radiation. These conditions are more intense and sustained during the
summer months in southern Taiwan.
Isoprene is a reactive hydrocarbon emitted predominantly by vegetation and plays a critical role
in atmospheric chemistry. Chaozhou, situated in an agriculturally active region with abundant
vegetation, provides a suitable landscape for biogenic isoprene emissions. The diurnal profiles clearly
show peak concentrations during late morning to early afternoon hours (10:00–15:00), aligning with
maximum solar radiation and leaf temperature, both of which are key drivers of isoprene synthesis and
emission in plants.
However, the presence of isoprene within a mixed species profile—rather than as a standalone
source—is surprising and suggests that other non-biogenic sources may also contribute. Our previous
studies on isoprene in less industrial areas and mostly urban in nature found very distinct diurnal features
with mixing ratios peaking at noon and decreasing to very low levels at night (Chang et al., 2014; Hsieh
et al., 2017). One plausible contributor is local vehicular activity around the station. The relatively
higher isoprene levels in summer may reflect the combined influence of both biogenic activity and
enhanced volatilization from road surfaces or tire materials.
The CPF analysis (Fig. S5c) further supports a local and regionally distributed source profile, with
increased conditional probabilities from the SSW and NW sectors during both seasons. These wind
directions correspond with vegetated and agricultural areas surrounding the site, reinforcing the role of
nearby land cover in influencing isoprene levels. However, the non-directional component and broader
sector coverage also hint at additional inputs from anthropogenic sources, particularly traffic-related
processes and diffuse sources. The seasonal and diurnal behaviors highlight the sensitivity of isoprene
to environmental drivers and the complex nature of its sources in mixed-use regions like Chaozhou.

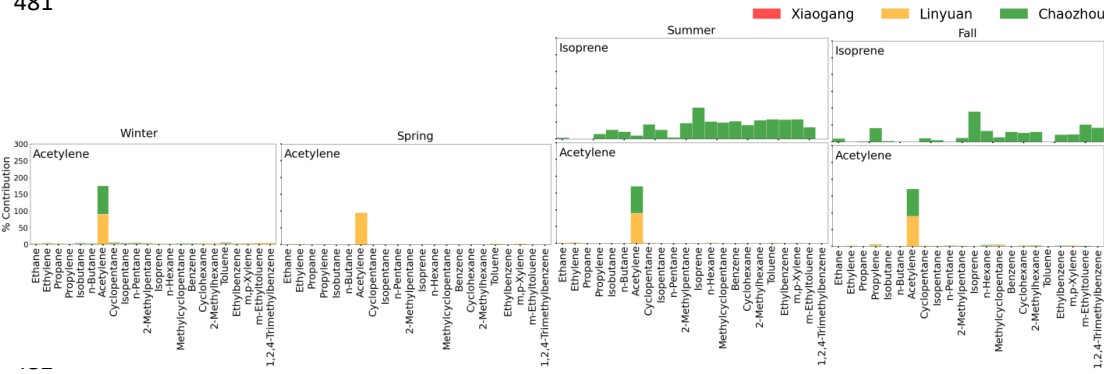

**Figure 5: Distinct source profiles of NMHC at the three sites in 2024**
*b) Acetylene factor*
Acetylene ($C_2H_2$) is a highly flammable hydrocarbon gas and is mainly used as a fuel gas for oxy-
acetylene welding, cutting, brazing, and soldering. On the other hand, the filling process for high-



pressure cylinders is also prone to leaking. These activities are a known source of acetylene in these
study regions. From the PMF results in Fig. 5, there is a resolved factor for the single species of
acetylene and no other accompanying species source at Linyuan and Chaozhou, suggesting that
acetylene is a pronounced local source, which is consistent with the prior knowledge of the acetylene
filling plant in the region.
PMF-resolved time series of the acetylene factor reveals a temporal pattern, with elevated
contributions during winter and fall, and generally lower levels in summer (Fig. S3). These variations
are likely driven by meteorological influences, such as enhanced atmospheric stability and reduced
boundary layer height in colder months, which favor the accumulation of locally emitted pollutants. In
contrast, stronger vertical mixing and photochemical degradation in summer likely contribute to the
overall reduction in acetylene levels during this period.
Complementary insights are provided by the CPF analysis, which highlights the directional
characteristics of acetylene sources. Seasonal CPF polar plots show that Linyuan (orange) consistently
exhibits strong directional signals, with the highest probabilities originating from the NW–NE across
all seasons. This directional consistency supports the presence of a persistent local source to the north
of the site–aligned with the known location of an acetylene filling plant in the region. The elevated CPF
values in these wind sectors reinforce the interpretation that the PMF-resolved acetylene factor reflects
a geographically localized emission source, with its observed variability driven more by meteorological
transport conditions than by changes in source activity. In contrast, no acetylene factor was resolved at
Xiaogang, and this absence is consistent with its prevailing wind patterns. Seasonal wind rose data show
that Xiaogang is predominantly influenced by W and NW winds (Fig. S6)—directions that do not align
with the position of the acetylene source near Linyuan, which lies to the southeast of Xiaogang. As a
result, the site remains largely unaffected by emissions from the filling facility.
While the PMF analysis resolved an acetylene factor at both Linyuan and Chaozhou, the
contribution at Chaozhou was substantially lower (Fig.S3). This disparity suggests that, unlike
Linyuan—where a known acetylene filling plant serves as a prominent and consistent local source—
Chaozhou is likely influenced by more diffuse or intermittent combustion-related activities. These may
include small-scale combustion activities such as residential biomass burning, occasional waste
incineration, or localized activities like welding. The absence of a strong directional signal in the CPF
analysis for Chaozhou further supports the interpretation of a non-point or variable origin of acetylene
in this area.

**3.3 PMF-Trigger back trajectory integration**

It is noteworthy that an acetylene-related factor was clearly resolved in the PMF results, especially
for Linyuan but not for Xiaogang. This outcome aligns well with the raw PAMS observations, where
acetylene concentrations at Linyuan are markedly elevated, often exhibiting sharp spikes reaching
several hundred ppb (Fig. S1), especially in winter. In contrast, levels at Xiaogang remain consistently
low, rarely exceeding 20 ppb. The resolution of the acetylene factor at Linyuan can be attributed to the
high signal-to-noise ratio in the PAMS data, where sharp and frequent concentration spikes provided a



clear signal for PMF to distinguish this source from others, resulting in a well-defined temporal profile
that closely matched the observed data. This supports the reliability of PMF in capturing source-specific
signatures when driven by strong observational input.
Beyond factor profiles and species contributions, the PMF-resolved acetylene factor at Linyuan
also resulted in a well-defined temporal profile that closely matched the observed data. Figure 6
demonstrates this consent with a high correlation coefficient ($R^2$) of over 0.99. This high level of
agreement underscores the ability of PMF to cleanly resolve the profile dominated by a single species,
acetylene, and to accurately reproduce the temporal variability of pollutant levels. While many PMF
studies focus primarily on profile interpretation, this study demonstrates that time series validation can
provide an additional, rigorous layer of confidence in the factor identification—highlighting the
robustness of the PAMS dataset and the reliability of the PMF analysis. As a result, acetylene serves as
an intrinsic reference species, providing an internal check on the PMF analysis.
Because the acetylene events at Linyuan are extremely distinct, a triggered back-trajectory
analysis was conducted to investigate their source locations. This analysis used observational data from
the monitoring station as input, under the assumption that the local wind field was representative of the
broader surrounding area. The model was configured to calculate air parcel trajectories 15 minutes
backward from the observation site. Geographic Information System (GIS) tools and Google Maps API
were used to spatially visualize the trajectories and identify potential emission hotspots through
trajectory receptor pattern overlay analysis. Setting an appropriate concentration threshold is critical for
isolating representative pollution events. If the threshold is too low, resulting trajectories may be overly
dispersed, making source identification difficult. Conversely, a threshold set too high may highlight
only extreme events, which may not represent typical source behavior. This analysis applied a threshold
of 96.85 ppb (95th quantile) to ensure that only significant acetylene events were considered.
The overlay of multiple high-concentration trajectories consistently pointed to an area near the
acetylene filling facility (Fig. 6). Although the backward trajectories do not align perfectly with the
suspected emission source, this deviation is likely due to the influence of complex coastal meteorology
that can affect low-level air parcel paths, especially under transitional wind conditions. Moreover, the
trajectory model may not fully capture local turbulence and terrain effects, contributing to the observed
offset. Despite this, the temporal patterns of peaked acetylene, combined with the site's position relative
to the dominant wind direction, support the likely influence of the identified source. These findings
demonstrate the value of high-resolution PAMS data in capturing pollutant events and reinforce the
consistency between observational measurements and PMF-based source apportionment. Finally, the
back-trajectory method, particularly when applied in a triggered mode during elevated events, offers
enhanced spatial resolution and source identification capabilities that complement and extend beyond
PMF results.





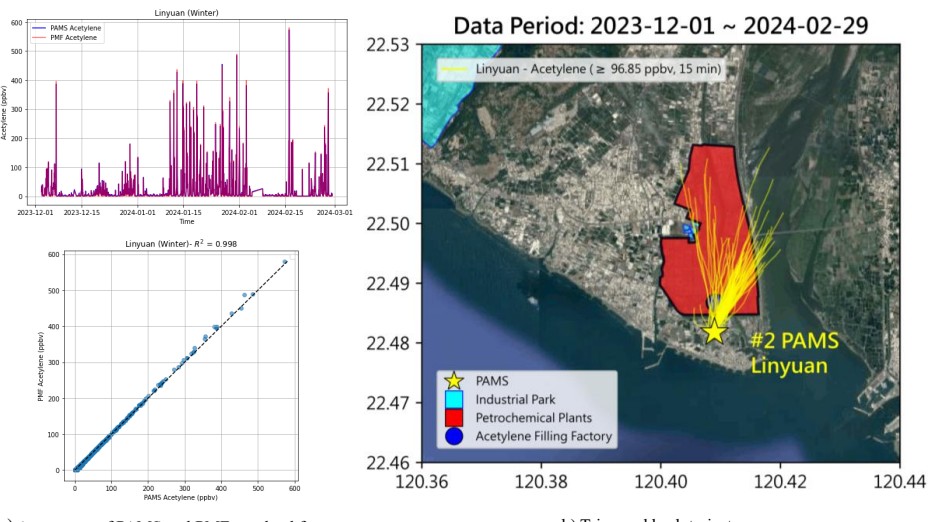

a) Agreement of PAMS and PMF-resolved factor          b) Triggered back trajectory

**Figure 6: The acetylene factor at Linyuan, (a) time series comparison and (b) Triggered back trajectory analysis of spike levels. Base map from Google Maps (Map data ©2025 Google).**

**3.4 Quantitative estimates of source contributions.**

The differences in source contribution across the three sites are not arbitrary but rather reflect the distinct roles each monitoring station plays within the regional emission landscape. Linyuan functions as a source site because it is home to major refinery and petrochemical facilities. Xiaogang, by contrast, presents a mixed urban-industrial environment, while Chaozhou acts as a downwind receptor site in a predominantly rural setting. Since many NMHC species originate from multiple source types, the resolved factors were grouped into broader categories: petroleum industry (including petro I & II, refinery, and gasoline evaporation), mixed, aged air mass, acetylene, and biogenic.

Figure 7 illustrates the seasonal contribution of these grouped sources. Interestingly, the petroleum-related source contribution is most prominent (33-71%), especially at Xiaogang, despite Linyuan being the location of core petrochemical activities. This apparent contradiction can be explained by local meteorological conditions—particularly prevailing winds and sea–land breeze effects—which frequently transport emissions from the surrounding industrial corridor toward Xiaogang. These wind-driven dynamics create a receptor–source relationship, wherein Xiaogang accumulates both locally emitted and regionally transported pollutants. In contrast, Linyuan shows a strong signal from the acetylene factor, pointing to localized activities with distinct point-source characteristics. Meanwhile, Chaozhou's elevated aged air mass contribution underscores its role as a downwind receptor site.

Aged air masses consistently influence all sites and seasons, with contributions ranging from 12% to 30%. Chaozhou, in particular, exhibits higher contributions—up to 30% in spring and above 27% in winter and fall, while summer shows the lowest levels. Its inland, rural settings with limited local industrial activity suggest that the site primarily receives aged air masses transported from upwind regions. Supporting this interpretation, evidence from CPF analysis (Fig. 4) and Fig. S4 highlight the




role of local recirculation driven by sea-land breeze, which enhances the aging of air masses near the site.
Additionally, long-range transport under prevailing winter monsoon winds further reinforces the elevated
aged air signal observed at Chaozhou. At the same time, Xiaogang emerged as the second contributor of
aged air mass, with notable peaks in winter, spring, and fall. As a result, the three sites with their unique
source-receptor characteristics produce very dynamic source apportionment results that vary in season
and locations.

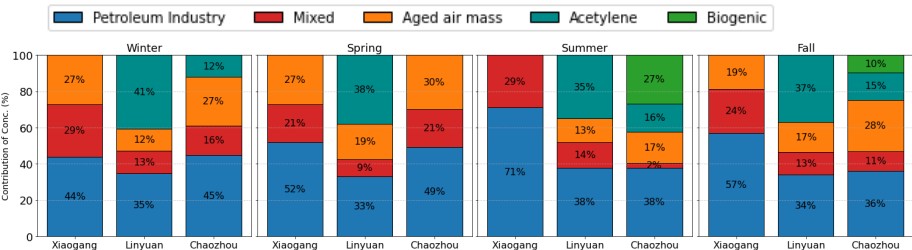


**Figure 7: Seasonal contributions of grouped source factors at each site. The details of the eight**
**resolved factors are provided in Fig. S7**
**3.5 OFP dynamics under varying ozone conditions**

595       Given that photochemical reactions predominantly occur during daylight hours, the OFP was

calculated exclusively for daytime periods, defined using sunrise and sunset times, derived from
astronomical data tailored to each station's geographic locations and observation dates. Seasonally
averaged OFP values attributed to source factors were highest at Xiaogang ($113.20 \pm 23.60$ µg/m³),
followed by Linyuan ($102.73 \pm 40.93$ µg/m³), and lowest at Chaozhou ($65.38 \pm 9.00$ µg/m³), as shown
in Fig. S8. Spatially, the distribution of OFP contributions varied consistently with their previously
identified roles. Notably, while petroleum-related sources exhibited the highest contributions in terms
of concentration, they were not always the leading contributors to OFP. For example, at Xiaogang, the
petroleum factor ranked second in OFP, overtaken by the mixed source factor (Fig. S8). This is primarily
due to the presence of highly reactive species—such as aromatics—in the mixed source profile, which
significantly elevates its OFP despite relatively lower concentrations. In Xiaogang, the mixed source
was the dominant contributor to OFP during most seasons, with petroleum-related sources only
surpassing it in summer. To further explore source contributions under varying ozone pollution
conditions, the dataset was classified based on the maximum daily 8-hour average (MDA8) ozone
concentrations: days with MDA8 larger than 60 ppb were designated as pollution (POL) days, and those
with MDA8 between 40 and 60 ppb as moderate pollution (MOD) days. OFP was recalculated
accordingly, and the percentage contributions of each source factor are presented in Figure 8. The
comparison between POL and MOD days reveals notable shifts in source influence across the region.



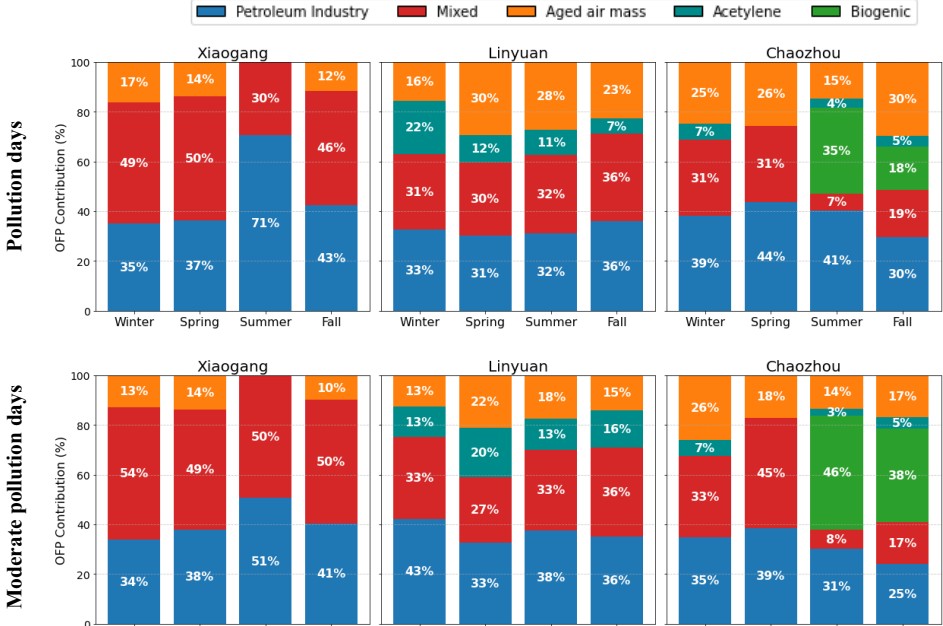

**Figure 8: Corresponding daytime OFP during pollution and moderate pollution days of ozone**

Mixed and petroleum-related sources consistently contribute a substantial portion of OFP under both conditions. For example, mixed sources are the dominant contributor, particularly at urban-industrial sites such as Xiaogang, suggesting a more dominant role in ozone precursor formation even under less intense ozone conditions. Meanwhile, petroleum-related sources increase their contribution, particularly at Linyuan, indicating their stronger association with moderate ozone levels. Generally, large-scale industrial and traffic emissions (petro-, mixed, and acetylene combined) provide a more reactive precursor condition prone to ozone formation in Xiaogang and Linyuan areas, consistent with their dense industrial landscape.

Meanwhile, the aged air mass factor also exerted a notable influence, especially on ozone pollution days, particularly during ozone pollution days and at downwind sites like Linyuan and Chaozhou. This points to the role of regional transport and atmospheric aging in shaping local ozone levels. The elevated OFP from aged air mass under polluted days suggests that many reactive NMHCs driving ozone formation are not emitted locally, but transported from upwind areas after undergoing photochemical processing. Importantly, aged air masses may also carry ozone itself, effectively raising the background ozone concentration and providing a higher baseline upon which local photochemical production builds (Nguyen et al., 2025). This dual role—delivering both reactive precursors and ozone—can intensify pollution episodes.

Notably, at Chaozhou, biogenic emissions mark a significant contribution to the OFP, particularly during the summer and fall seasons. This is largely attributed to the release of isoprene from its extensive agriculture and vegetation landscape. The transported NOx can trigger ozone formation from rapid photochemical reaction with biogenic isoprene. The impact of biogenic emissions was particularly





evident during moderate ozone pollution days, suggesting that moderate ozone episodes appear to be
more sensitive to the combined effects of natural emissions, transported NOx, and local atmospheric
chemistry. This underscores the importance of considering seasonal and spatial dynamics in emission
control strategies, particularly in areas like Xiaogang and Linyuan, where both anthropogenic and
biogenic sources may interact synergistically to elevate ozone levels.
Regarding chemical speciation, the dominant contributors to OFP across all sites and seasons were
aromatics, alkanes, and alkenes, due to their high photochemical reactivity. Aromatics—particularly
toluene and m,p-xylene (Fig. S9)—accounted for 32–61% of the OFP during moderate pollution days,
with the highest contributions observed at Xiaogang. Alkanes, primarily ethane, contributed 20–35%,
with similar levels at Xiaogang and Linyuan, and slightly higher at Chaozhou during pollution days.
Alkenes, especially propylene, contributed 15–24%, with the highest at Xiaogang during pollution days
and at Linyuan during moderate pollution days. Additional contributions were observed from ethyne—
notably at Linyuan—and from isoprene at Chaozhou, where they played a more prominent role during
moderate pollution days. Although acetylene is relatively low in MIR, the sheer tonnage of emissions
predominantly from the filling plant still leads to significant OPF in the downwind area. The persistence
of higher OFP contributions during moderate ozone days indicates that precursor control should not be
limited to pollution events alone. Thus, Effective ozone management requires a multi-faceted approach
that considers anthropogenic and biogenic sources, seasonal variability, and transportation influences to
design more adaptive, location-specific mitigation policies.
**4. Conclusion**
This study provides a comprehensive analysis of NMHC concentrations and their sources across
three distinct sites in southern Taiwan: Linyuan, Xiaogang, and Chaozhou. The findings revealed
significant spatial heterogeneity in NMHC concentrations and source profiles, reflecting the diverse
land use and industrial activities across the region. Linyuan, a dense industrial landscape, exhibited the
highest average NMHC concentrations, followed by the mixed urban-industrial environment of
Xiaogang, with the lowest levels observed at the predominantly agricultural site of Chaozhou. PMF
analysis identified eight distinct source factors contributing to NMHCs at the three sites: petrochemical
I & II, refinery, gasoline evaporation, mixed (vehicular/solvent), aged air mass, acetylene, and biogenic.
The strength of this study lies in the use of PMF-resolved time series output, allowing for the
identification of concentration spikes indicative of episodic emission events. This output reached a high
level of consent with the PAMS data ($R^2$ over 0.99), which triggered targeted back-trajectory analyses.
These consistently traced emissions to a nearby acetylene filling facility north of the Linyuan industrial
area. The integration of PMF-resolved time series data with trajectory modeling reinforces the
credibility of the source apportionment results and underscores the high quality and temporal resolution
of the observational data. It enabled a more precise attribution of pollution sources and facilitated the
isolation and examination of individual pollution events.
In addition, this study explored the dynamics of OFP. They were calculated specifically for
daytime periods—when photochemical reactions are most active. Seasonally averaged OFP was highest



at Xiaogang (113.20 ± 23.60 µg/m³), followed by Linyuan (102.73 ± 40.93 µg/m³), and lowest at the
downwind rural site Chaozhou (65.38 ± 9.00 µg/m³). While petroleum-related sources contributed the
highest concentrations of NMHCs, the mixed source factor—which includes highly reactive species
such as aromatics—often contributed more to OFP, particularly at Xiaogang. These findings emphasize
that reactivity plays a critical role in ozone formation and should be considered alongside emission
volume. Across ozone pollution levels, petroleum and mixed sources remained dominant, but their
influence shifted: mixed sources were more important during moderate ozone days at urban-industrial
sites, while petroleum sources dominated in Linyuan under similar conditions. This pattern may reflect
early impacts of emission controls, more evident under high pollution but less so on moderate days.
Given their higher occurrence, moderate pollution episodes still offer valuable insights for identifying
key sources relevant to ozone formation. This study demonstrates a refined source apportionment
approach using multi-site, year-round, high-frequency NMHC measurements, each characterized by a
distinct source–receptor relationship. This approach provides a more comprehensive spatiotemporal
representation and yields deeper insights into ozone pollution management in the region.
**Data availability:** All raw data can be provided by the corresponding authors upon request.
**Author contribution:** JLW and CHW formed the conceptualization; HCH developed the model code
and performed the modeling; DHN and HCH analyzed the data; DHN wrote the original draft; JLW,
NHL, CHW, MCL, and DHN reviewed and edited the manuscript.
**Competing interests:** The authors declare that they have no conflict of interest.

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
