# Peer review of "Leveraging PMF Time Series Characteristics from Multi-PAMS Measurements for"

_EGUsphere, 2025_

## Author Comment (AC1)

**Point-to-point responses**

Manuscript title: Leveraging PMF Time Series Characteristics from Multi-PAMS Measurements for NMHC Source Investigation and Ozone Formation Insights

Authors: Nguyen et al.

Summary

This manuscript presents a comprehensive source apportionment study of non-methane hydrocarbons (NMHCs) in Taiwan's Kaoping region using high-resolution year-round measurements from three Photochemical Assessment Monitoring Stations (PAMS). Positive Matrix Factorization (PMF) was applied to resolve eight distinct NMHC source factors, complemented by Conditional Probability Function (CPF) analysis and a novel "triggered backtrajectory" approach based on episodic peaks in PMF time series. The study finds that petroleum and mixed (vehicular/solvent) sources dominate NMHC contributions and ozone formation potential (OFP), with significant influence even during moderate ozone days.

The manuscript is timely, methodologically innovative, and policy-relevant. It provides strong observational evidence and demonstrates a refined framework for linking receptor modeling with spatial source attribution. However, there are some improvements necessary regarding methodological uncertainty and interpretive depth, which should be addressed before publication. This amounts to a minor revision.

**Response:** On behalf of the co-authors, we appreciate your constructive comments on improving our manuscript. All the comments have been read through, and responses have been made accordingly. The detailed responses are as follows. Revisions made to the manuscript in response to these comments are indicated in Red.

**1. Dependence on PMF and CPF assumptions**

o The analysis relies on PMF, which is sensitive to input selection, number of factors, and uncertainty estimates. While stability tests are mentioned, the details of these test are not given and unclear what test are done (e.g., factor rotations, influence of excluded low-S/N species). This would strengthen confidence in the robustness of source identification.

**Response:** In this study, uncertainty estimation followed the U.S. EPA PMF 5.0 protocol, incorporating species-specific analytical error fractions and MDL-based rules (below MDL: ½ MDL concentration,  $\frac{5}{6}$  MDL uncertainty). Low S/N species (with stricter ratio S/N < 0.1 instead of 0.2) were excluded, while weak species  $(0.1 \le \text{S/N} < 2)$  were down-weighted to prevent instability from low-quality inputs. The stability of PMF solutions was examined through multiple diagnostics, including 100-run random seed tests for each factor number (from 3 to 8), evaluation of Q(robust)/Q(true) ratios, and interpretability checks. Additional Bootstrap (BS) tests were conducted to assess factor stability. The selected eight-factor solution showed consistent profiles reproduced in over 95% of bootstrap runs with Q(robust)/Q(true) ratios that is close to 1, confirming the

robustness of the source identification. The related text in the method section has been revised to enhance readability. Please see lines: 201-216

"In this study, we applied the U.S. EPA's PMF 5.0 software to perform source apportionment of NMHCs measured at the three PAMS sites in the Kaoping region. The input to the model consisted of concentration and uncertainty matrices constructed from hourly NMHC measurements. Uncertainty (Uij) was calculated based on species concentrations (Xij) and method detection limits (MDL) as follows:

$$U_{ij} = \sqrt{(0.5 \times MDL_j)^2 + (error fraction \times X_{ij})^2}$$
(1)

For concentrations below MDLs, the value was substituted with ½ MDL with uncertainty set at 5 MDL. Missing values were excluded from the input dataset to maintain model reliability. Species selection was based on signal-to-noise (S/N) ratios and detection frequency. Specifically, species with S/N ratios < 0.1, while those with 0.1 ≤ S/N < 2 were down-weighted as weak species. This screening process resulted in a final set of 22 out of the 54 measured species used for modeling. Model stability was assessed through multiple diagnostic procedures. First, factor numbers from 3 to 8 were tested, each with 100 independent runs using random seed initialization. The optimal number of factors was selected based on Q(robust)/Q(true) values that close to 1.0, reproducibility of factor profiles across runs using Bootstrap (BS) analyses (>95% matching), and the interpretability and physical plausibility of the resulting source profiles. These stability tests collectively help confirm the factor solution, providing a robust and well-constrained representation of NMHC sources in the study region."

o The model stability of the CPF could be visualized.

Response: Thank you for your constructive comment. While the 0.75 quantile is commonly used and generally robust for CPF analysis at urban or industrial sites (Pekney et al., 2006; Chen et al., 2019; Huang & Hsieh, 2019; Wu et al., 2024), our sensitivity tests indicated that it was not the optimal threshold in our case. Using one full year of hourly NMHC observations, we compared CPF results across several percentile thresholds, from 0.60 to 0.85. Acetylene was used as a reference factor because its potential source direction—originating from the northern area—is well established, providing a reliable basis for evaluating CPF stability. The CPF patterns remained relatively consistent between the 0.60 and 0.70 quantiles but began to degrade at higher thresholds. Specifically, CPF values dropped sharply and directional features became less meaningful above the 0.75 quantile, as the number of high-percentile events per wind sector decreased substantially, reducing statistical reliability. We therefore selected the 0.70 quantile as an optimal solution. This threshold effectively filters out moderate events, focuses on high-impact

episodes, and retains sufficient samples per sector for robust and statistically stable CPF estimation. Although higher thresholds (0.75–0.80) emphasize more episodic or localized plumes, in our dataset, they provided too few valid samples for reliable directional inference. The 0.70 quantile thus balances reliability and selectivity while preserving the major CPF directional features relevant for interpretation. The CPF method section was accordingly revised to clarify this selection (see lines 228-229), and the sensitivity test results are provided in Table S1 of the Supplementary Materials.

"CPF was computed as the ratio of the number of times the factor contributions exceeded the threshold within a given wind sector  $(n_{\Delta\theta})$  to the total number of valid observations in that sector  $(m_{\Delta\theta})$ . Wind direction was divided into 16 equal intervals (22.5° per sector) to ensure robust analysis. A 70th-percentile threshold was adopted to isolate plume events, as it effectively filters out moderate events while retaining sufficient data for statistically stable and interpretable CPF results (Table S1). Higher CPF values in specific wind sectors indicated stronger contributions from sources in that direction. CPF plots were generated for each PMF factor to visualize dominant source directions and assess consistency with known emission source locations, meteorological patterns, and local topography."

"Table S1 Sensitivity of CPF directional patterns to percentile thresholds. Acetylene, whose source direction is well established as primarily originating from the northern industrial area, was used as a reference factor for this evaluation. CPF patterns were consistent between the 0.60 and 0.70 quantiles but became unstable above 0.75, as higher thresholds yielded fewer valid samples per sector and less coherent directional features. Therefore, the 0.70 quantile was determined to be the most appropriate solution, effectively filtering out moderate events while retaining sufficient data for statistically stable and interpretable CPF results."

| Wind_sector | CPF-0.6 | CPF-0.65 | CPF-0.7 | CPF-0.75 | CPF-0.8 | CPF-0.85 |
|-------------|---------|----------|---------|----------|---------|----------|
| 1           | 1       | 1        | 1       | 1        | 1       | 0.895    |
| 2           | 0.803   | 0.799    | 0.799   | 0.747    | 0.631   | 0.486    |
| 3           | 0.379   | 0.379    | 0.379   | 0.268    | 0.172   | 0.111    |
| 4           | 0.202   | 0.191    | 0.169   | 0.135    | 0.079   | 0.034    |
| 5           | 0.140   | 0.116    | 0.093   | 0.093    | 0.093   | 0.047    |
| 6           | 0.233   | 0.178    | 0.068   | 0.041    | 0.027   | 0.014    |
| 7           | 0.287   | 0.207    | 0.069   | 0.023    | 0.011   | 0.011    |
| 8           | 0.319   | 0.181    | 0.026   | 0        | 0       | 0        |
| 9           | 0.595   | 0.214    | 0.048   | 0        | 0       | 0        |
| 10          | 0.308   | 0.192    | 0.077   | 0.077    | 0.038   | 0.038    |
| 11          | 0.367   | 0.306    | 0.245   | 0.143    | 0.122   | 0.082    |
| 12          | 0.181   | 0.111    | 0.056   | 0.028    | 0.014   | 0.014    |
| 13          | 0.170   | 0.112    | 0.076   | 0.054    | 0.036   | 0.022    |

| 14 | 0.545 | 0.545 | 0.545 | 0.500 | 0.455 | 0.364 |
|----|-------|-------|-------|-------|-------|-------|
| 15 | 0.900 | 0.900 | 0.900 | 0.900 | 0.800 | 0.700 |

o The inability to fully separate vehicular and solvent sources is acknowledged. However, this limitation has important implications for regulatory application and should be discussed more explicitly in the context of control strategies.

**Response:** We have expanded the discussion in the OFP section to address more about the implications of incomplete separation between vehicular and solvent sources. Please see lines 671-683.

"Under moderate-ozone conditions, the mixed-source factor contributes the largest share of OFP. These episodes are frequently associated with lower wind speeds and reduced mixing heights (AACOG, 2015), favoring the accumulation of locally emitted species from traffic, solvent, and light industrial activities. The coexistence of these emissions provides a balanced supply of reactive aromatics and olefins, sustaining ozone production. This pattern reflects the VOC-limited photochemical environment prevalent in southern Taiwan, where ozone formation is more sensitive to reactive VOCs than to NOx levels (Chang et al., 2022), emphasizing that moderate-ozone episodes are primarily governed by local accumulation and NMHC composition. However, the mixed-source factor reflects overlapping characteristics of vehicular and solvent-related species. This incomplete separation introduces some uncertainty in source-specific attribution of OFP. Nevertheless, this overlap mirrors the reality of urban environments. From a regulatory perspective, this finding suggests that ozone control measures targeting only traffic or solvent emissions in isolation may be insufficient. Effective mitigation in southern Taiwan, therefore, requires coordinated management of both mobile and solvent-related species."

**2. Treatment of Uncertainty**

Uncertainty quantification appears limited. For example, details of the OFP estimates are not shown or explained and the specific chemistry assumptions are not mentioned. Needs to be added.

**Response:** We thank the reviewer for pointing this out. In the revised manuscript, we have added a description of the OFP estimation method, including the chemical mechanism, and a description of the uncertainty. Please see lines: 244-261.

**"2.5 Ozone formation potential and uncertainty consideration**

The OFP of each NMHC species was estimated using the Maximum Incremental Reactivity (MIR) coefficients developed by Carter (2010). The OFP for compound i was calculated as:

$$OFP_i = C_i \times MIR_i \tag{3}$$

Where  $C_i$  (ppb) is the measured mixing ratio of the species, and  $MIR_i$  (g O3 g-1 VOC) is its reactivity coefficient. The MIR scale represents ozone yield under low VOC/NOx (i.e., high-NOx or VOC-limited) conditions, where ozone formation is primarily sensitive

to changes in VOC abundance. This assumption aligns with previous photochemical studies indicating that ozone formation in southern Taiwan is predominantly VOC-limited (Chang et al., 2022).

The uncertainty associated with OFP estimation was inherently accounted for during the PMF analysis. Each NMHC species was assigned an uncertainty value based on its measured concentration, method detection limit, and error fraction (10%). Values below the MDL were replaced with ½ MDL, and the corresponding uncertainty was set to ½ MDL, while missing values were excluded. These uncertainties were used to construct the PMF input uncertainty matrix, which determines the weighting of each data point in the model fitting. As the OFP was calculated from the PMF-resolved factor contribution time series, the measurement uncertainties are inherently reflected in the factor contributions. Therefore, it ensures that the OFP estimates incorporate the measurement uncertainty structure already considered."

**3. Interpretation of Moderate Ozone Days**

The finding that mixed sources dominate OFP under moderate ozone conditions is highly relevant, but the mechanistic explanation is underdeveloped. Are these results consistent with VOC-limited regimes? How do meteorological conditions (e.g., mixing height) shape these patterns? Expanding the discussion would enhance both scientific and policy relevance.

**Response:** We thank the reviewer for this valuable comment. We have enhanced the discussion of OFP patterns under moderate-ozone conditions (Section 3.6, lines 663-683). The revised text now provides a clearer mechanistic explanation linking the dominance of mixed sources to local photochemical regimes and meteorological influences. Specifically, we added the following points:

a-Mechanistic explanation of photochemical regimes under moderate ozone days: Southern Taiwan, particularly Kaohsiung, is characterized by a VOC-limited ozone formation regime (Chang et al., 2022). Under such conditions, ozone production is more sensitive to changes in VOC reactivity than in NOx. Therefore, the dominance of mixed sources—which combine traffic, solvent, and industrial VOCs with moderate-to-high MIR values—during moderate ozone episodes is consistent with this chemistry. Small increases in reactive VOCs from these sources can effectively enhance ozone production.

**b-Meteorological conditions:**

Moderate ozone days typically occur under weak horizontal transport and low-to-moderate mixing heights (AACOG, 2015), leading to the accumulation of local VOCs. As a result, the lower boundary layer and limited ventilation during moderate ozone days enhance the influence of locally mixed sources with highly reactive species on OFP.

c-Implications for control strategy:

These findings suggest that under typical VOC-limited conditions, reductions in mixed-source emissions—particularly aromatics and light alkenes—could yield more effective

ozone mitigation during local accumulation events, whereas regional transport events would require coordinated upwind emission control.

"Mixed and petroleum-related sources consistently contribute a substantial portion of OFP under both conditions. For example, mixed sources are the dominant contributor, particularly at urban-industrial sites such as Xiaogang, underscoring their strong role in ozone precursor formation even during less intense ozone conditions. Meanwhile, petroleum-related sources showed an enhanced contribution at Linyuan, suggesting a stronger association with moderate ozone levels. Overall, large-scale industrial and traffic-related emissions (petro-, mixed, and acetylene-related) provide highly reactive precursor conditions conducive to ozone formation across Xiaogang and Linyuan areas, consistent with their dense industrial landscape.

Under moderate-ozone conditions, the mixed-source factor contributes the largest share of OFP. These episodes are frequently associated with lower wind speeds and reduced mixing heights (AACOG, 2015), favoring the accumulation of locally emitted species from traffic, solvents, and light industrial activities. The coexistence of these emissions provides a balanced supply of reactive aromatics and olefins, sustaining ozone production. This pattern reflects the VOC-limited photochemical environment prevalent in southern Taiwan, where ozone formation is more sensitive to reactive VOCs than to NOx levels (Chang et al., 2022), emphasizing that moderate-ozone episodes are primarily governed by local accumulation and NMHC composition. However, the mixed-source factor reflects overlapping characteristics of vehicular and solvent-related species. This incomplete separation introduces some uncertainty in source-specific attribution of OFP. Nevertheless, this overlap mirrors the reality of urban environments. From a regulatory perspective, this finding suggests that ozone control measures targeting only traffic or solvent emissions in isolation may be insufficient. Effective mitigation in southern Taiwan, therefore, requires coordinated management of both mobile and solvent-related species."

**Response:** As the discussion on OFP is further enhanced, the related text in the conclusion has been revised accordingly. Please see lines: 732-754

"In addition, this study explored the dynamics of OFP. They were calculated specifically for daytime periods—when photochemical activity is most pronounced. Seasonally averaged OFP was highest at Xiaogang (113.20  $\pm$  23.60  $\mu g/m^3$ ), followed by Linyuan (102.73  $\pm$  40.93  $\mu g/m^3$ ), and lowest at the downwind rural site Chaozhou (65.38  $\pm$  9.00  $\mu g/m^3$ ). Although petroleum-related sources contributed the largest fraction of NMHC concentrations, the mixed source factor—enriched in highly reactive species such as aromatics—often dominates the OFP, particularly at Xiaogang. Under moderate-ozone conditions (MDA8 40–60 ppb), the factor became the principal driver of ozone formation, consistent with local accumulation under stagnant meteorological conditions and limited

vertical mixing. The coexistence of reactive aromatics and olefins within this factor sustained ozone production, reflecting a VOC-limited regime typical of southern Taiwan. Across ozone pollution levels, petroleum and mixed sources remained dominant, but their relative influence varied with site characteristics. Mixed sources exerted stronger effects during moderate-ozone episodes at urban–industrial locations, whereas petroleum-related sources dominated in Linyuan under similar conditions. These results suggest that frequent, moderate-ozone episodes are primarily driven by locally accumulated reactive NMHCs. This pattern may also reflect the early impacts of emission control measures, which are more effective under high-pollution conditions but less so during moderate episodes. Given their higher occurrence, moderate pollution episodes still offer valuable insights into the interplay between local accumulation, VOC reactivity, and emission composition that governs ozone formation in the region.

Overall, this study demonstrates the strength of a refined source apportionment approach using multi-site, year-round, high-frequency NMHC measurements, each reflecting distinct source—receptor dynamics. The findings offer a more comprehensive spatiotemporal understanding of ozone formation mechanisms and provide a scientific basis for coordinated control of mobile, solvent-related, and petroleum-associated emissions across southern Taiwan."

**Technical Comments**

**1. Abstract and Summary**

o The abstract is long and technical. Consider reducing methodological detail in favor of emphasizing findings and policy implications.

**Response:** We appreciate this suggestion. The abstract has been revised significantly as follows.

"Ozone pollution remains a persistent challenge in Taiwan's Kaoping region, driven by dense industrial and urban emissions. To better constrain source-specific non-methane hydrocarbons (NMHCs), key ozone precursors, this study advances receptor-based model Positive Matrix Factorization (PMF) with year-long high-resolution measurements from three Photochemical Assessment Monitoring Stations (PAMS). By integrating multi-site observations over a full annual cycle, this framework captures spatial contrasts and enhances the interpretation of emission source characteristics and their contributions to ozone formation. Eight distinct NMHC sources were resolved, with petroleum-related emissions contributing the largest across all sites. The model effectively captured source-specific signatures, notably the acetylene factor at Linyuan (R2 = 0.99 with observations), serving as an intrinsic check on PMF performance. Aged air mass factors were found to play a meaningful role in ozone pollution events, particularly at downwind receptors,

underscoring the influence of atmospheric aging and regional transport. Meanwhile, mixed sources from vehicular and solvent activities, though smaller in mass, contributed disproportionately to ozone formation potential (OFP) due to their high reactivity. Seasonal and pollution-level analyses further indicate that even under moderate ozone conditions (MDA8 40–60 ppb), local mixed sources remained the principal OFP contributors in a VOC-limited regime, where the buildup of reactive species maintains ozone production. This study provides the first regionally integrated, year-long PMF–OFP framework in southern Taiwan, offering a novel approach to link source-resolved VOC characteristics with photochemical ozone formation under varying pollution regimes."

**2. Figures**

o Some figures (e.g., Fig. 3, Fig. S3) are very information-heavy. Simplifying or providing summary schematics could aid readability.

**Response:** We acknowledge the reviewer's concern. However, these figures present essential information that cannot be further simplified without losing key analytical details. To enhance readability and self-explanation, we have refined the figure captions to help readers better interpret the results.

"Figure 3: Summary of common source profiles of NMHC at the three sites in 2024. This figure presents the percentage contribution of six common source factors (Petro I, Petro II, Refinery, Gasoline evaporation, Mixed, and Aged air mass) to the total NMHC burden at three monitoring sites: Xiaogang (red), Linyuan (orange), and Chaozhou (green) across four seasons (Winter, Spring, Summer, and Fall). Each panel represents a specific source factor and season combination. The stacked bars within each panel show the relative contribution of NMHC species to that source factor. Consistent source profiles of fingerprint species were observed across seasons and at all three monitoring sites, underscoring the robustness of the PMF results."

"Fig. S3 Seasonal variation of daily mean contributions of NMHCs from PMF-resolved factors. This figure illustrates the daily variability of identified source factors (Petro I, Petro II, Refinery, Gasoline evaporation, Mixed, Aged air mass, and Acetylene) over the course of a year at three monitoring sites: Xiaogang (red), Linyuan (orange), and Chaozhou (green). To interpret this figure, follow the trends of the lines for each site within each season to observe how a specific source factor changes over time and reveal episodic emission events that indicate the changes in source activity. Shaded columns represent weekends. Any difference in these shades can potentially indicate the impact of human activities."

o S7 legend not readable, S5a vertical axis units are missing.

**Response:** There were errors in these figures, and corrections have been made. The legend in Figure S7 has been updated for clarity, and a vertical axis label has been added to Figure S5a.

**Fig. S5** Temporal variation and CPF analysis for biogenic factors over the seasons, a) Time series of daily average factor contribution, b) Diurnal variation, and c) CPF analysis

**Fig. S7** Contribution of concentration from eight resolved source factors. Xiaogang tends to have a relatively balanced influence among mixed, refinery, and aged air mass sources, suggesting complex source-receptor dynamics at this site. This reflects Xiaogang's transitional setting between industrial clusters and transportation hubs, where port activity, on-road emissions, and solvent-use industries intersect, producing overlapping chemical signals captured under the mixed factor.

**3. Literature Context**

o While many relevant studies are cited, the discussion could further situate the work in the context of ozone regime sensitivity studies (VOC- vs. NOx-limited conditions), which are critical for interpreting the results.

**Response:** We have added text to further situate the work in the context of ozone regime sensitivity studies (VOC- vs. NOx-limited conditions), which aligns with the interpretation of our OFP results. Please see lines: 53-65

"The impact of VOCs on ozone formation critically depends on the local chemical regime—whether ozone production is VOC-limited or NOx-limited (Sillman, 1999; Kleinman et al., 2002). In VOC-limited environments, common in densely industrialized and urbanized areas, ozone levels are more responsive to changes in reactive VOC concentrations, whereas in NOx-limited conditions, ozone formation is constrained by nitrogen oxide availability. Several studies in East and Southeast Asia have emphasized this spatial heterogeneity of ozone sensitivity (Li et al., 2019; Wang et al., 2021; Ren & Xie, 2022). In Taiwan, both modeling and observational evidence indicate that southern and western regions typically exhibit VOC-limited or transition regimes, while rural and downwind areas are more NOx-limited (Chen et al., 2021; Chang et al., 2022). This regime dependence underscores the need for region-specific precursor management and highlights the importance of identifying the dominant reactive VOC sources that most effectively drive ozone formation. Understanding these sensitivities provides an essential framework for interpreting ozone formation potential (OFP) derived from NMHC source contributions."

Recommendation: Minor revision

This is a strong and innovative paper that makes a meaningful contribution to atmospheric chemistry and air quality management. However, addressing the above issues—particularly uncertainty analysis, interpretation regarding ozone formation under moderate conditions, and clearer presentation of policy-relevant results—would further strengthen the manuscript.

**Response:** Thank you for your positive assessment of our work. We appreciate your insightful comments and have addressed all of your concerns with revisions to the manuscript. We believe these changes have significantly strengthened the manuscript.

**References**

- AACOG. (2015). Conceptual Model, Ozone Analysis of the San Antonio Region, Updates through Year 2014, (Ed.) N.R.T. Department, Texas Commission on Environmental Quality. San Antonio, Texas
- Carter, W.P. (2010). Development of the SAPRC-07 chemical mechanism. *Atmospheric Environment*. 44(40), 5324-5335. https://doi.org/10.1016/j.atmosenv.2010.01.026
- Chang, J.H.-W., Griffith, S.M., Kong, S.S.-K., Chuang, M.-T., Lin, N.-H. (2022). Development of a CMAQ-PMF-based composite index for prescribing an effective ozone abatement strategy: A case study of sensitivity of surface ozone to precursor VOC species in southern Taiwan. *Atmospheric Chemistry Physics Discussions*. 2022, 1-48.
- Chen, C.-H., Chuang, Y.-C., Hsieh, C.-C., Lee, C.-S. (2019). VOC characteristics and source apportionment at a PAMS site near an industrial complex in central Taiwan. *Atmospheric Pollution Research*. 10(4), 1060-1074. 10.1016/j.apr.2019.01.014
- Chen, S.-P., Liu, W.-T., Hsieh, H.-C., Wang, J.-L. (2021). Taiwan ozone trend in response to reduced domestic precursors and perennial transboundary influence. *Environmental Pollution*. 289, 117883. <a href="https://doi.org/10.1016/j.envpol.2021.117883">https://doi.org/10.1016/j.envpol.2021.117883</a>
- Huang, Y.S., Hsieh, C.C. (2019). Ambient volatile organic compound presence in the highly urbanized city: source apportionment and emission position. *Atmospheric Environment*. 206, 45-59. 10.1016/j.atmosenv.2019.02.046
- Kleinman, L.I., Daum, P.H., Lee, Y.N., Nunnermacker, L.J., Springston, S.R., Weinstein-Lloyd, J., Rudolph, J. (2002). Ozone production efficiency in an urban area. *Journal of Geophysical Research:* Atmospheres. 107(D23), ACH 23-1-ACH 23-12. <a href="https://doi.org/10.1029/2002JD002529">https://doi.org/10.1029/2002JD002529</a>
- Li, K., Jacob, D.J., Liao, H., Shen, L., Zhang, Q., Bates, K.H. (2019). Anthropogenic drivers of 2013–2017 trends in summer surface ozone in China. *Proceedings of the National Academy of Sciences*. pp. 422-427.
- Pekney, N.J., Davidson, C.I., Zhou, L., Hopke, P.K. (2006). Application of PSCF and CPF to PMF-modeled sources of PM2. 5 in Pittsburgh. *Aerosol Science Technology*. 40(10), 952-961.
- Ren, J., Xie, S. (2022). Diagnosing ozone-NOx-VOC sensitivity and revealing causes of ozone increases in China based on 2013–2021 satellite retrievals. *Atmospheric Chemistry Physics Discussions*. 2022, 1-22. <a href="https://doi.org/10.5194/acp-22-15035-2022">https://doi.org/10.5194/acp-22-15035-2022</a>
- Sillman, S. (1999). The relation between ozone, NOx and hydrocarbons in urban and polluted rural environments. *Atmospheric environment*. 33(12), 1821-1845. https://doi.org/10.1016/S1352-2310(98)00345-8
- Wang, W., van der A, R., Ding, J., van Weele, M., Cheng, T. (2021). Spatial and temporal changes of the ozone sensitivity in China based on satellite and ground-based observations.

- *Atmospheric Chemistry Physics.* 21(9), 7253-7269. <a href="https://doi.org/10.5194/acp-21-7253-2021">https://doi.org/10.5194/acp-21-7253-2021</a>
- Wu, S., Alaimo, C.P., Green, P.G., Young, T.M., Zhao, Y., Liu, S., Kuwayama, T., Kleeman, M.J. (2024). Source apportionment of Volatile Organic Compounds (VOCs) in the South Coast Air Basin (SoCAB) During RECAP-CA. *Atmospheric Environment*. 338, 120847.